# Bacterial precursors and unsaturated long-chain fatty acids are biomarkers of North-Atlantic deep-sea demosponges

**Anna de Kluijver**[1]*, **Klaas G. J. Nierop**[1]*, **Teresa M. Morganti**[2], **Martijn C. Bart**[3], **Beate M. Slaby**[4], **Ulrike Hanz**[5], **Jasper M. de Goeij**[3], **Furu Mienis**[5], **Jack J. Middelburg**[1]

**1** Department of Earth Sciences, Faculty of Geosciences, Utrecht University, Utrecht, Netherlands, **2** Max Planck Institute for Marine Microbiology, Bremen, Germany, **3** Department of Freshwater and Marine Ecology, Institute for Biodiversity and Ecosystem Dynamics, University of Amsterdam, Amsterdam, Netherlands, **4** GEOMAR Helmholtz Centre for Ocean Research Kiel, Kiel, Germany, **5** NIOZ-Royal Netherlands Institute for Sea Research and Utrecht University, Den Burg, Texel, Netherlands

* a.dekluijver@uu.nl, anna.dekluijver@rvo.nl (ADK); k.g.j.nierop@uu.nl (KGJN)

**Data Availability Statement:** All relevant data are within the manuscript and its Supporting Information files.

## Abstract

Sponges produce distinct fatty acids (FAs) that (potentially) can be used as chemotaxonomic and ecological biomarkers to study endosymbiont-host interactions and the functional ecology of sponges. Here, we present FA profiles of five common habitat-building deep-sea sponges (class Demospongiae, order Tetractinellida), which are classified as high microbial abundance (HMA) species. *Geodia hentscheli*, *G. parva*, *G. atlantica*, *G. barretti*, and *Stelletta rhaphidiophora* were collected from boreal and Arctic sponge grounds in the North-Atlantic Ocean. Bacterial FAs dominated in all five species and particularly isomeric mixtures of mid-chain branched FAs (MBFAs, 8- and 9-Me-$C_{16:0}$ and 10- and 11-Me-$C_{18:0}$) were found in high abundance (together $\geq$ 20% of total FAs) aside more common bacterial markers. In addition, the sponges produced long-chain linear, mid- and *a(i)*-branched unsaturated FAs (LCFAs) with a chain length of 24–28 C atoms and had predominantly the typical $\Delta^{5,9}$ unsaturation, although the $\Delta^{9,19}$ and (yet undescribed) $\Delta^{11,21}$ unsaturations were also identified. *G. parva* and *S. rhaphidiophora* each produced distinct LCFAs, while *G. atlantica*, *G. barretti*, and *G. hentscheli* produced similar LCFAs, but in different ratios. The different bacterial precursors varied in carbon isotopic composition ($\delta^{13}C$), with MBFAs being more enriched compared to other bacterial (linear and *a(i)*-branched) FAs. We propose biosynthetic pathways for different LCFAs from their bacterial precursors, that are consistent with small isotopic differences found in LCFAs. Indeed, FA profiles of deep-sea sponges can serve as chemotaxonomic markers and support the concept that sponges acquire building blocks from their endosymbiotic bacteria.

## Introduction

Sponges are abundant inhabitants of nearly all aquatic ecosystems including the deep-sea [1]. They are sessile filter feeders with unique features, such as their enormous filtration capacity

**Funding:** This research has been performed in the scope of the EU SponGES project, which received funding from the European Union's Horizon 2020 research and innovation programme under grant agreement No. 679849. This document reflects only the authors' views and the Executive Agency for Small and Medium-sized Enterprises (EASME) is not responsible for any use that may be made of the information it contains. Further support included ERC starting grant agreement No. 715513 to Dr. J. M. de Goeij and the Netherlands Earth System Science Center to Prof. J. J. Middelburg. Dr. F. Mienis is supported by the Innovational Research Incentives Scheme of the Netherlands Organisation for Scientific Research (NWO-VIDI grant no. 0.16.161.360). Additional funds come from the DFG Cluster of Excellence "The Ocean in the Earth System" at the University of Bremen (grant. 49926684) and from the ERC Adv Grant ABYSS, both to Prof. Antje Boetius (grant no. 294757). The funders had no role in study design, data collection and analysis, decision to publish, or preparation of the manuscript.

**Competing interests:** The authors have declared that no competing interests exist.

and their symbiosis with dense and diverse communities of (sponge-specific) microbes (algae, bacteria, archaea) [2,3] that contribute to their ability to thrive at nearly all depths and latitudes. The endosymbionts, which can occupy >50% of sponge volume [4], serve as energy source for sponges and provide a diverse pallet of metabolites and metabolic pathways that are beneficial to the sponge (reviewed in [2]). A prominent class of metabolites produced by the sponge and its endosymbionts are lipids. Lipid analysis of sponges started in the 1970s [5,6] and was sparked by the diversity and unique structures of fatty acids (FAs), of which extensive reviews exist [7–9]. Characteristic of sponges is the presence of unusual poly-unsaturated, long chain (≥24 carbons(C)) FAs (LCFAs), with a typical $\Delta^{5,9}$ unsaturation (named "demospongic acids", because of their first discovery in demosponges [5,10]). These LCFAs constitute a major part of sponge membrane phospholipids (PLs) and probably serve a structural and functional role [11]. Sponges, because of their endosymbionts, are rich in bacterial FAs with high diversity, including not only the common *iso* (*i*) and *anteiso* (*a*)-branched FAs, but also more unusual ones. Typical of demosponges are a high abundance of mid-chain branched FAs (MBFAs), that are thought to be produced by sponge-specific eubacteria [12], and a presence of branched LCFAs [12,13]. As branching is assumed to be introduced by microbes and not by the sponge host, the presence of branched LCFAs provides information on biosynthetic interactions between endosymbionts and host [12,14]. Monoenic FA, e.g. $C_{16:1}\omega7$, abundant in bacteria [15], have been identified as precursors for LCFAs with ω7 configuration [16]. Accordingly, the position of unsaturation also provides insight in bacteria-host biosynthetic interactions.

In addition, sponge FA composition may have taxonomic value, at least on a higher classification level (e.g. class level), since Demospongiae, Hexactinellida ('glass' sponges), Calcarea, and Homoscleromorpha have distinct FA profiles [17]. However, the chemotaxonomic value on a lower classification level is disputable, since composition may alter with environmental conditions [18]. The FA composition of sponges, especially combined with (natural abundance) stable isotope analysis, has been shown a valuable tool to infer dietary information on sponges, such as feeding on coral mucus [19], phytoplankton [20] and methane-fixing endosymbionts [21].

The North-Atlantic Ocean is home to extensive sponge grounds, that are widespread along the continental shelves, seamounts, and on the abyssal plains [22,23]. *Geodiidae* and other sponge species of order Tetractinellida (class Demospongiae) are major constituents of these sponge grounds, representing >99% of sponge ground benthic biomass [23–25]. *Geodiidae* spp. are high microbial abundance (HMA) sponges that harbor rich, diverse and specific microbial communities (bacteria and archaea) involved in several biogeochemical processes, as observed in *G. barretti* [26]. This is reflected in the FA composition of *G. barretti* that is dominated by bacterial FAs [12], including the distinct MBFAs that represent 28% of total FAs [12]. However, the FA profiles of other *Geodiidae* are not described in the literature yet.

In this study we analyzed the FA profiles of five common deep-sea Tetractinellids, from different assemblages distinguished by temperature in the North Atlantic: the Arctic sponge ground assemblages accommodate *G. parva*, *G. hentscheli*, and *Stelletta* spp. (e.g. *S. rhaphidiophora*) dwelling at temperatures below 3–4°C, and the boreal assemblages accommodate *G. barretti* and *G. atlantica* amongst others, which are typically found at temperatures above 3°C [23,27]. Based on the chemical configuration and the presence of branching in LCFAs, we propose biosynthetic pathways and show that these are consistent with the C isotope ($\delta^{13}C$) signatures of LCFAs and bacterial precursors. The high abundance of endosymbiont markers that are precursors of LCFAs, indicate that these deep-sea sponges use their endosymbionts as metabolic source.

## Methods

### Sponge collection

Common habitat-building sponges of class Demospongiae, order Tetractinellida, were collected in the North-Atlantic Ocean by remotely operated vehicle (ROV) and box cores during different scientific expeditions. *G. atlantica (n* = 2) specimens were collected on the Sula Reef between 266–295 m depth during an expedition in August 2017 with the Norwegian research vessel G.O. Sars (64˚42'N 7˚59'E). *G. barretti (n* = 6) individuals were obtained from the Barents Sea (70˚47N 18˚03'E) around 300 m water depth on a subsequent G.O. Sars expedition in August 2018 [28]. Norwegian research expeditions do not require special permits for sample collection in this region. During the same expedition, *G. hentscheli*, *G. parva*, *Stelletta rhaphidiophora* (all *n* = 1) were collected at 550–600 m depth on the summit of Schulz Bank (73˚50′ N, 7˚34′ E) [29]. *G. hentscheli* (*n* = 3), *G. parva* (*n* = 3), and *S. rhaphidiophora* (*n* = 2) specimens were retrieved on an Arctic expedition with the German research vessel Polarstern (AWI Expedition PS101) in September–October 2016 at 690–1000 m depth from Langseth Ridge, located in the permanently ice-covered Central Arctic (from 87˚N, 62˚E to 85˚55'N 57˚45'E). Sample collection in international water does not require special permits. Sponges collected during the G.O. Sars expeditions were immediately dissected on board and sponges collected from Langseth Ridge were frozen at -20˚C and dissected (frozen) in the lab. Subsamples (*n* = 3) from random parts of individual sponges were freeze-dried, grinded to obtain a fine powder. The powdered subsamples of sponges from Schulz Bank and Barents Sea were mixed to obtain a species representative sample, while a subsample of the interior of sponges was analyzed in case of Langseth Ridge specimens. The voucher specimens from Langseth Ridge are stored at the University of Bergen, Norway. Voucher specimens of *G. barretti* and *G. atlantica* are stored at University of Amsterdam, Netherlands and the voucher specimens from Schulz Bank are stored at the Netherlands Institute for Sea Research, Texel, the Netherlands.

### Lipid extraction and FAME preparation

Approximately 100 mg of sponge powder of each individual sponge was used per extraction. Sponge lipids were extracted with a modified Bligh and Dyer protocol [30], which was developed at NIOZ Yerseke [31–33]. We adjusted this protocol by replacing chloroform with dichloromethane (DCM), because of lower toxicity. The whole protocol is available online: dx.doi.org/10.17504/protocols.io.bhnpj5dn. In short, sponge tissue samples were extracted in a solvent mixture (15 mL methanol, 7.5 mL DCM and 6 mL phosphate (P)-buffer (pH 7–8)) on a roller table for at least 3 hours. Layer separation was achieved by adding 7.5 mL DCM and 7.5 mL P-buffer. The DCM layer was collected, and the remaining solution was washed a second time with DCM. The combined DCM fraction was evaporated to obtain the total lipid extract (TLE), which was subsequently weighed. An aliquot of the TLE was separated into different polarity classes over an activated silica column. The TLE was first eluted with 7 mL DCM (neutral lipids), followed by 7 mL acetone (glycolipids) and 15 mL methanol (phospholipids). The phospholipid (PL) fraction, which was used for further analysis, was converted into fatty acid methyl esters (FAMEs) using alkaline methylation (using sodium methoxide in methanol with known $\delta^{13}$C). Alkaline methylation is recommended for complex lipid mixtures [34]. After methylation, FAMEs were collected in hexane and concentrated to ~100 μL hexane for gas chromatography (GC) analysis.

For this study, two individual sponge samples per species were selected for detailed analysis. Aliquots of the FAME samples were used for double bond identification using dimethyl disulfide (DMDS) derivatization [35]. Samples reacted overnight at 40˚C in 50 μL hexane, 50 μL

DMDS and 10 μL 60 mg/mL $I_2$. The reaction was stopped by adding 200 μL hexane and 200 μL $Na_2S_2O_3$. The hexane layer was collected, and the aqueous phase was washed twice with hexane. The combined hexane fraction was dried, subsequently eluted over a small $Na_2SO_4$ column using in DCM: methanol (9:1) to remove any water and re-dissolved in hexane in a GC-vial for GC-analysis. Another aliquot of FAME sample was used for methyl-branching identification using catalytic hydrogenation with Adams catalyst ($PtO_2$) and hydrogen. Each FAME sample, dissolved in ~3 mL ethyl acetate with 10–30 mg $PtO_2$ and a drop of acetic acid, was bubbled with hydrogen gas for at least 1 h, after which the reaction vial was closed, and stirred overnight at room temperature. Subsequently, each sample was purified over a small column consisting of $MgSO_4$ (bottom) and $Na_2CO_3$ (top) using DCM and analyzed after re-dissolving it in ethyl acetate.

### FAME analysis

FAMEs were analyzed on a gas chromatograph (GC) with flame ionization detector (FID) (HP 6890 series) for concentrations and GC-mass spectrometry (MS) (Finnigan Trace GC Ultra) for identification on a non-polar analytical column (Agilent, CP-Sil5 CB; 25 m x 0.32 mm x 0.12 μm). Samples were injected cold-on-column. The GC oven was programmed from 70–130˚C at 20˚C/min and subsequently at 4˚C/min to 320˚C, at which it was hold for 20 min. The GC–FID was operated at a constant pressure of 100 kPa, whereas the GC–MS was operated at a constant flow of 2.0 mL min$^{-1}$. The MS was operated in Full Data Acquisition mode, scanning ions from $m/z$ 50–800. The $^{13}C/^{12}C$ isotope ratios of individual FAMEs were determined by analyzing samples in duplicate on a GC-combustion-isotope ratio mass spectrometer (IRMS) consisting of a HP 6890N GC (Hewlett-Packard) connected to a Delta-Plus XP IRMS via a type-III combustion interface (Thermo Finnigan), using identical GC column and settings as for GC-MS.

Retention times were converted to equivalent chain length (ecl) based on the retention times of $C_{12:0}$, $C_{16:0}$, and $C_{19:0}$ FAMEs. $C_{19:0}$ FAME was also used to quantify the concentrations of individual FAMEs (μg g$^{-1}$ DW) [36]. The $δ^{13}C$ values obtained by GC-C-IRMS were corrected for the added C atom of the methylation agent. The data were analyzed and plotted in R [37] with R-package RLims [36].

## Results

The lipid yield of *G. barretti*, *G. hentscheli*, *G parva*, and *S. rhaphidiophora* was similar, around 2–3% of dry weight (DW). Only *G. atlantica* had a lower lipid yield, about 1.6% of DW. The PL derived FA (PLFA) profiles of PL resembled those of TLE and the majority of FA seemed to be present in PL (S1 Table, [38]). The estimated total PLFA content was 6.7 ± 6.3 mg g$^{-1}$ DW (0.7%) ($n$ = 16, across all species). Identification was more difficult using TLEs, because LCFAs co-eluted with sterols, hence PLFA chromatography was used for identification and composition analysis.

### Identification

Chemical structures of individual FAs were identified by retention times (ecl), interpretation of their mass spectra and/or by identification using a NIST library. The assignments were verified with reference mixtures (bacterial and general FA mixtures from Sigma Aldrich) and by literature comparison (e.g. the reference ecl lists from NIOZ Yerseke [31]).

FAs are presented in both ω and Δ (IUPAC) annotation to avoid unambiguity and in a hybrid form, which is typical of sponge LCFA annotation [17,39] (Table 1). Unsaturation is described as $C_{x:y}$, where x is the number of C atoms and y is the number of double bonds,

**Table 1. Fatty Acid (FA) composition in % of total PLFA of deep-sea demosponge species (order Tetractinellida):** *Geodia atlantica* (*Ga*), *G. barretti* (*Gb*), *G. hentscheli* (*Gh*), *G. parva* (*Gp*) and *Stelletta rhaphidiophora* (*Sr*). FA names are given in ω and Δ notation and a hybrid form, with corresponding total C atoms (C) and equivalent chain length (ecl). FA categories match with those of Fig 2. Only FAs with abundance ≥1% (in at least one species) are shown. Numbers in bold are ≥ 10% of total FAs.

| | | | | Species | *Ga* | *Gb* | *Gh* | *Gp* | *Sr* |
|---|---|---|---|---|---|---|---|---|---|
| | FA notation | | | N | 2 | 6 | 4 | 4 | 3 |
| Ecl | FA (ω) | FA (Δ) | C | Category | \multicolumn FA composition (%) | | | | |
| 13.68 | $C_{14:0}$ | $C_{14:0}$ | 14 | Other | 0.9 | 1.0 | 1.0 | 0.8 | 1.5 |
| 14.17 | $Me\text{-}C_{14:0}$ | $Me\text{-}C_{14:0}$ | 15 | Bacteria | 0.7 | 1.4 | 1.4 | 1.3 | 1.8 |
| 14.38 | $i\text{-}C_{15:0}$ | $13\text{-}Me\text{-}C_{14:0}$ | 15 | | 3.0 | 3.5 | 3.1 | 2.5 | 4.0 |
| 14.46 | $a\text{-}C_{15:0}$ | $12\text{-}Me\text{-}C_{14:0}$ | 15 | | 2.6 | 2.6 | 2.2 | 2.0 | 4.3 |
| 15.35 | $Me\text{-}C_{15:0}$ | $Me\text{-}C_{15:0}$ | 16 | | 0.6 | 0.9 | 1.6 | 1.8 | 1.6 |
| 15.59 | $C_{16:1}\omega9$ | $C_{16:1}\Delta^{7}$ | 16 | | 1.6 | 0.5 | 2.6 | 3.3 | 2.5 |
| 15.68 | $C_{16:1}\omega7$ | $C_{16:1}\Delta^{9}$ | 16 | | 6.3 | 8.7 | 7.8 | 6.3 | 8.2 |
| 15.78 | $C_{16:1}\omega5$ | $C_{16:1}\Delta^{11}$ | 16 | | 1.5 | 2.1 | 1.9 | 1.6 | 2.7 |
| 16 | $C_{16:0}$ | $C_{16:0}$ | 16 | Other | 5.7 | 3.7 | 3.5 | 3.2 | 4.2 |
| 16.31 | $i\text{-}C_{17:1}\omega7$ | $15\text{-}Me\text{-}C_{16:1}\Delta^{9}$ | 17 | Bacteria | 4.2 | 5.4 | 1.4 | 1.0 | 1.4 |
| 16.45 | 8- and 9-$Me\text{-}C_{16:0}$ | 8- and 9-$Me\text{-}C_{16:0}$ | 17 | | 8.3 | **10** | **14** | **11** | **13** |
| 16.62 | $i\text{-}C_{17:0}$ | $15\text{-}Me\text{-}C_{16:0}$ | 17 | | 1.1 | 1.3 | 1.2 | 1.3 | 2.2 |
| 16.68 | $a\text{-}C_{17:0}$ | $14\text{-}Me\text{-}C_{16:0}$ | 17 | | 1.4 | 1.3 | 0.9 | 0.8 | 1.2 |
| 17.40 | $Me\text{-}C_{17:0}$ | $Me\text{-}C_{17:0}$ | 18 | | 3.1 | 2.8 | 1.7 | 1.9 | 2.0 |
| 17.65 | $C_{18:1}\omega9$ | $C_{18:1}\Delta^{9}$ | 18 | | 1.2 | 0.2 | 1.5 | 1.4 | 1.5 |
| 17.72 | $C_{18:1}\omega7$ | $C_{18:1}\Delta^{11}$ | 18 | | 3.1 | 4.2 | 3.8 | 3.9 | 3.7 |
| 18 | $C_{18:0}$ | $C_{18:0}$ | 18 | Other | 4.6 | 3.7 | 3.0 | 3.0 | 3.1 |
| 18.11 | $Me\text{-}C_{18:1}\omega12$ or $\omega14$ | $Me\text{-}C_{18:1}\Delta^{6}$ or $\Delta^{4}$ | 19 | Bacteria | 2.0 | 2.9 | 4.0 | 4.4 | 4.9 |
| 18.46 | 10- and 11-$Me\text{-}C_{18:0}$ | 10- and 11-$Me\text{-}C_{18:0}$ | 19 | | **12** | **17** | **20** | **23** | **19** |
| 18.78 | $cy\text{-}C_{19:0}$ | $cy\text{-}C_{19:0}$ | 19 | | 1.0 | 1.2 | 1.2 | 0.8 | 1.3 |
| 20.85 | $C_{22:6}\omega3$ | $C_{22:6}\Delta^{4,7,10,13,16,19}$ | 22 | Other | 1.8 | 1.3 | 0.2 | 0.6 | |
| 23.17 | $C_{24:2}\Delta^{5,9}$ ($\omega15$) | $C_{24:2}\Delta^{5,9}$ | 24 | Sponge | | | | 1.2 | |
| 23.67 | $i\text{-}C_{25:2}\Delta^{5,9}$ ($\omega15$) | $23\text{-}Me\text{-}C_{24:2}\Delta^{5,9}$ | 25 | | | | | **11** | |
| 23.74 | $a\text{-}C_{25:2}\Delta^{5,9}$ ($\omega15$) | $22\text{-}Me\text{-}C_{24:2}\Delta^{5,9}$ | 25 | | | | | 4.2 | |
| 23.84 | $i\text{-}C_{25:1}\omega7$ | $23\text{-}Me\text{-}C_{24:1}\Delta^{17}$ | 25 | | 2.4 | 1.8 | | | |
| 24.73 | $C_{26:2}\Delta^{5,9}$ ($\omega17$) | $C_{26:2}\Delta^{5,9}$ | 26 | | 2.4 | 2.4 | 5.4 | 0.4 | 0.4 |
| 24.81 | $C_{26:2}\Delta^{9,19}$ ($\omega7$) | $C_{26:2}\Delta^{9,19}$ | 26 | | 4.3 | 8.4 | 8.5 | 0.6 | |
| 25.11 | $Me\text{-}C_{26:2}\Delta^{5,9}$ ($\omega17$) | $Me\text{-}C_{26:2}\Delta^{5,9}$ | 27 | | 9.4 | 4.5 | 1.3 | | 1.2 |
| 25.28 | $(a)i\text{-}C_{27:2}\Delta^{5,9}$ ($\omega7$) or $\Delta^{9,19}$ ($\omega17$) | 24 or 25-$Me\text{-}C_{26:2}\Delta^{5,9}$ or $\Delta^{9,19}$ | 27 | | 4.9 | 2.9 | 0.1 | | 0.7 |
| 25.96 | $C_{28:2}\Delta^{5,9}$ ($\omega21$) | $C_{28:2}\Delta^{5,9}$ | 28 | | | | | | 1.5 |
| 26.14 | $C_{28:2}\Delta^{11,21}$ ($\omega7$) | $C_{28:2}\Delta^{11,21}$ | 28 | | 1.9 | 1.8 | | 3.6 | |
| 26.71 | $Me\text{-}C_{28:2}\Delta^{5,9}$ ($\omega21$) | $Me\text{-}C_{28:2}\Delta^{5,9}$ | 28 | | | | | | 5.5 |

which is followed by Δ and all double bond positions from the carboxylic acid end in Δ notation, and the position of the first double bond from the methyl (terminal) end in ω notation (Table 1, Fig 1). Methyl branching according to IUPAC notation is described as y-Me-$C_x$, where y is the position of the branching from the carboxylic acid end and x is the number of C atoms at the backbone, excluding the branching (Fig 1). The ω notation follows the terminology of IUPAC for MBFAs, but deviates for terminally branched FAs. The penultimate (ω2) and pen-penultimate methyl branching (ω3) are described with ω notation as *iso* ($i\text{-}C_x$) and *anteiso* ($a\text{-}C_x$) where x is the total number of C atoms, including the branching (Table 1, Fig 1).

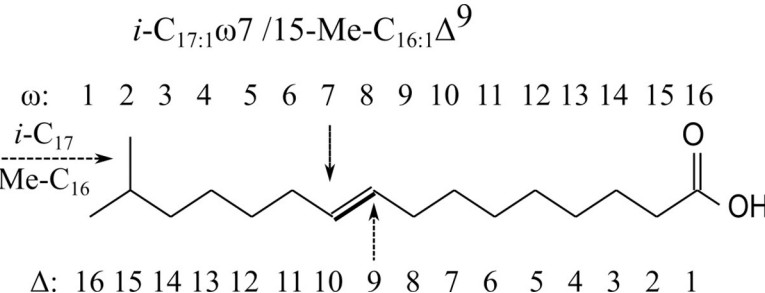

**Fig 1. Illustration of ω and Δ annotation for the chemical structure of *i*-C$_{17:1}$ω7 / 15-Me-C$_{16:1}$Δ$^9$.** The *a(i)* -notation for methyl branching describes the total number of C, while the Me-notation describes the number of C in the backbone. For sponge LCFAs, a mixture of both nomenclatures is, however, commonly used.

The elution order on an apolar column consists of FAMEs with methyl-branching close to the functional group to elute first, followed by the terminally (penultimate) branched *iso* (*i*, ω1) and pen-penultimate *anteiso* (*a*, ω2) FAMEs, and finally the unsaturated FAMEs, for which unsaturation closest to the functional group elutes first. Branched unsaturated FAMEs elute before branched straight FAMEs and straight FAMEs with the same C number elute last (Fig 2, Table 1).

The position of branching was also verified with MS spectra, as *i*-branching was characterized by a more intense [M$^+$-43] fragment ion and *a*-branching was characterized by an elevated fragment ion at [M$^+$-57]. The position of methyl branching in saturated MBFAs was identified via diagnostic mass fragments similar to [12]. The relative intensity of *m/z* 171, 185, 199 and [185+213] was used to identify the relative contributions of 8, 9, 10, and 11-Me branching, respectively (S1 Table). Because 11-Me-branching produces equal fragments of m/z 185 and 213, the excess of m/z 185 (213–185) was produced by 9-Me branching (S1 Table)

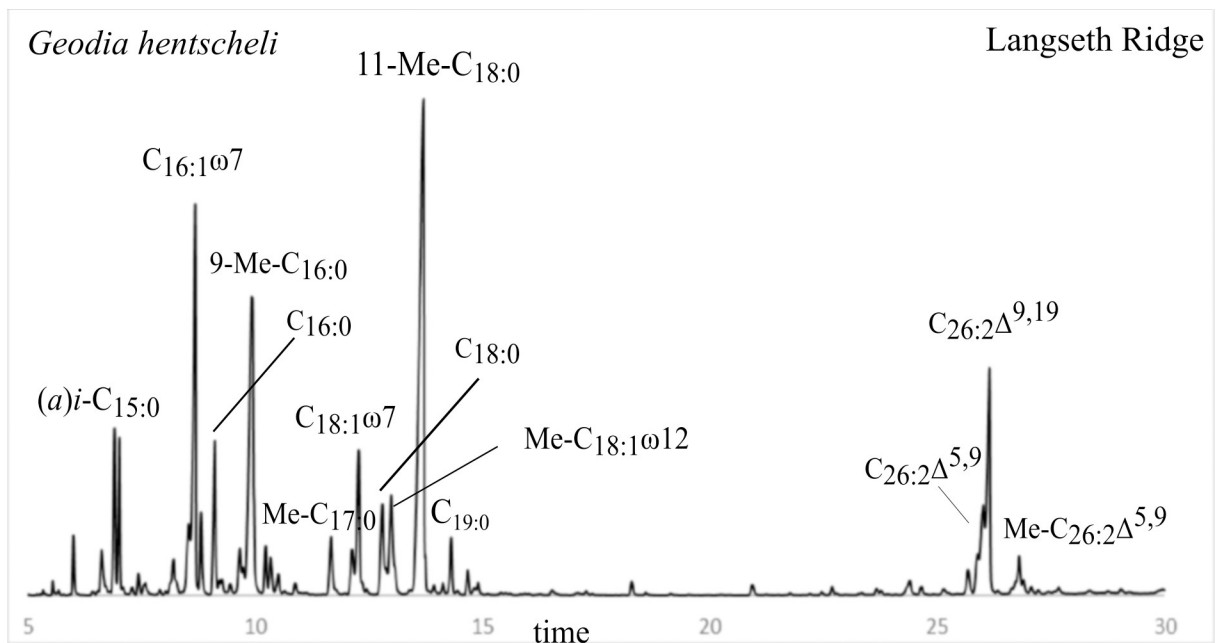

**Fig 2. GC trace of the FAME fraction extracted from demosponge *G. hentscheli* from Langseth Ridge (Central Arctic).** LCFA isomers often co-eluted or were at least not well separated as shown in this PLFA profile for C$_{26:2}$Δ$^{5,9}$ and C$_{26:2}$Δ$^{9,19}$.

[12]. The branching within unsaturated MBFAs was performed in hydrogenated samples, using similar diagnostic fragments and ecl of saturated FAMEs (S1 Table).

Identification of unsaturation positions was conducted after treatment with DMDS, which is straight-forward with mono-unsaturated FAMEs. However, for poly-unsaturated FAMEs, identification with DMDS becomes complicated, because of multiple possibilities for S(-Me) adducts. The $\Delta^{5,9}$ unsaturation, typical of sponge LCFAs, forms a cyclic thioether at the $C_6$ and $C_9$ position along with methylthio groups at $C_5$ and $C_{10}$ positions upon derivatization with DMDS. In addition, products are formed with either methylthio groups at $C_5$ and $C_6$ and a (unreacted) double bond at $C_9$ and $C_{10}$, and vice versa [40]. This has been useful for identifying the typical $\Delta^{5,9}$ configuration in sponges [41].When unsaturation is far apart, i.e. positions $\Delta^{9,19}$ and $\Delta^{11,21}$, both double bonds are converted to dimethyl disulfide adducts (S1 Fig for their mass spectra). Based on ecl and a combination of DMDS and hydrogenation, we identified branched-monoenic and dienic FAs.

## Fatty acid composition

The Arctic species (*G. hentscheli*, *G. parva*, *S. rhaphidiophora*) from Schulz Bank and Langseth Ridge had a similar PLFA profile (S1 Table), so we pooled the compositional data from the two locations (Table 1). The data are standardized to % of total PLFAs (hereafter FAs) to facilitate comparison, but actual FA concentrations ($\mu g\ g^{-1}$ DW) are available in S1 Table.

**Bacterial fatty acids.** Bacterial FAs, comprising branched and monoenic FAs with chain length $< C_{20}$, constituted the majority of total FAs in all five deep-sea demosponge species ($67 \pm 6\%$ mean $\pm$ SD of total FAs, used throughout text, $n = 19$, across all species) (Table 1, Fig 3) and can represent up to $79 \pm 2\%$ (in *S. rhaphidiophora*).

MBFAs dominated the FA profiles of all deep-sea demosponge species (Table 1, Fig 2), among them the most abundant were Me-$C_{18:0}$ (12–23%, Table 1), with branching at 9, 10, 11 with a predominance at position 11 (*m/z* [213+185]; 49% on average), followed by position 10

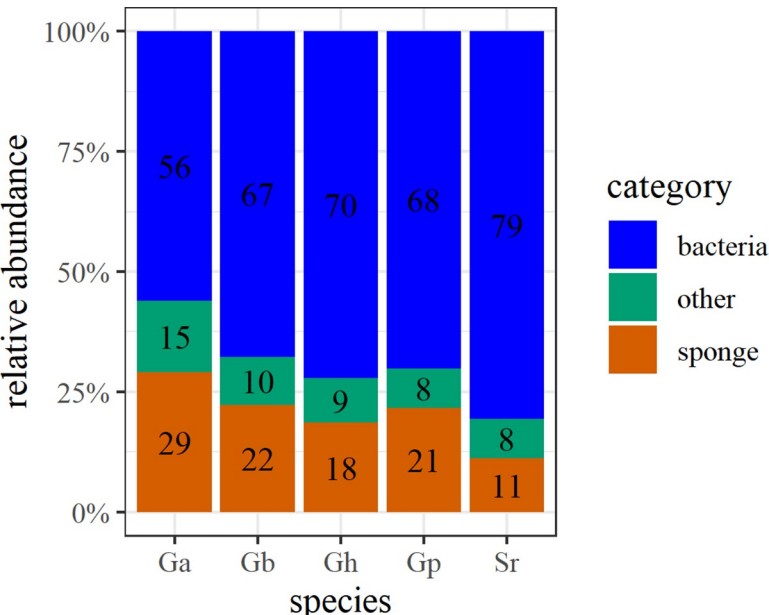

**Fig 3.** Average contribution of bacterial FAs (blue), sponge LCFAs (orange) and other FAs (green) to the total PLFAs of each species (abbreviated as in Table 1).

($m/z$ 199; 37% on average). The second most abundant FAs were Me-$C_{16:0}$ (8–14% of total FAs, Table 1), with branching at 8, 9, 10, 11 and a predominance of position 9 ($m/z$ 185; 36% on average) and 10 ($m/z$ 199; 33% on average). Also, Me-$C_{14:0}$, Me-$C_{15:0}$, and Me-$C_{17:0}$ were present, but in much lower abundance ($\leq$ 3% of total FAs for each, Table 1). Other branched (saturated) FAs found in all demosponges but less abundant, included $i$-$C_{15:0}$ (13-Me-$C_{14:0}$) and $a$-$C_{15:0}$ (12-Me-$C_{14:0}$), comprising 2–5% of total FAs for each, and $i$-$C_{17:0}$ (15-Me-$C_{16:0}$) and $a$-$C_{17:0}$ (14-Me-$C_{16:0}$), ranging from 1 to 2% of total FAs for each (Table 1).

Multiple monoenic FAs were found in the deep-sea demosponges. The most abundant were $C_{16:1}$ (ranging from 9% in *G. atlantica* to 14% in *G. parva* and *S. rhaphidiophora*), consisting of different isomers with the double bond at ω5, ω7, ω9, and ω11 positions. Isomers of $C_{18:1}$ with double bonds at ω7, ω8, ω9, ω11, ω12, ω13, ω14, and ω15 positions constituted 4–7% of total FAs. The ω7 unsaturation dominated in both $C_{16:1}$ and $C_{18:1}$ FAs. In addition, rare $C_{16:1}$ and $C_{18:1}$ FAs with methyl-branching were found in demosponges. The dominating unsaturation in $C_{16:1}$ was ω7 and the hydrogenated FAME sample indicated $i$-branching; $i$-$C_{17:1}$ω7 (15-Me-$_{16:1}\Delta^9$) represented 4–5% in boreal species (*G. barretti*, *G. atlantica*) and < 2% in Arctic species (*G. hentscheli*, *G. parva*, *S. rhaphidiophora*) (Fig 1, Table 1). The most abundant unsaturation in $C_{18:1}$ FAs was ω12 ($\Delta^6$) for *G. parva*, *G. hentscheli* and *S. rhaphidiophora* and ω14 ($\Delta^4$) for *G. barretti* and *G. atlantica*, and the Me group was in the middle of the chain, since no increased peaks for $i$- and $a$-$C_{19:0}$ were found in the corresponding hydrogenated fractions. The mid-Me branched $C_{18:1}$ isomers (Me-$C_{18:1}$ω4 and Me-$C_{18:1}$ω12) were found in all demosponge species and ranged between 2–5% (Table 1). Also, low amounts (< 1%) of $C_{15:1}$ and non-branched $C_{17:1}$ were found.

**Other fatty acids.** Linear FAs were predominantly $C_{14:0}$, $C_{16:0}$ and $C_{18:0}$ in all species (Table 1). Sponges contained only low amounts of FAs with a chain length between $C_{20}$ and $C_{24}$, such as $C_{20:5}$ω3 (< 1% in all species) and $C_{22:6}$ω3 (1.4 ± 0.9%, $n$ = 8) in boreal species *G. atlantica* and *G. barretti* and < 1% in Arctic species *G. hentscheli*, *G. parva* and *S. rhaphidiophora*.

**Sponge fatty acids.** LCFAs ($\geq$ $C_{24}$), typical of sponges, differed per species and consisted of 24–29 C atoms (Table 1). LCFAs represented 21 ± 6% ($n$ = 19, across all species), with the highest contribution (29%) in *G. atlantica* (Fig 3). The most common unsaturation in demosponges was $\Delta^{5,9}$, but also unsaturation at $\Delta^{9,19}$ and $\Delta^{11,21}$ was observed.

- **$C_{25}$**: The dominant LCFA in *G. parva* was 23-Me-$C_{24:2}\Delta^{5,9}$ ($i$-$C_{25:2}\Delta^{5,9}$), followed by 22-Me-$C_{24:2}\Delta^{5,9}$ ($a$-$C_{25:2}\Delta^{5,9}$), making up 15 ± 1% of total FAs. Isomers 23-Me-$C_{24:1}\Delta^{17}$ ($i$-$C_{25:1}$ω7) and (mid-)Me-$C_{24:1}\Delta^{17}$ (Me-$C_{24:1}$ω7) were present in boreal species (*G. atlantica* and *G. barretti*) representing together 2 ± 0.6% (Table 1).

- **$C_{26}$**: The dominant LCFA in *G. hentscheli* was $C_{26:2}\Delta^{9,19}$, followed by $C_{26:2}\Delta^{5,9}$, together they represented 14 ± 6% of total FAs in that species. *G. barretti* and *G. atlantica* also synthesized $C_{26:2}\Delta^{5,9}$ and $C_{26:2}\Delta^{9,19}$ in comparable amounts, representing together 11 ± 1% in *G. barretti* and 7% in *G. atlantica*. Trace amounts (<1%) of $C_{26:2}\Delta^{5,9}$ were present in *G. parva* and *S. rhaphidiophora*. Similarly, trace amount of $C_{26:2}\Delta^{11,21}$ (<1%) was found in *G. parva*.

- **$C_{27}$**: (mid-)Me-$C_{26:2}\Delta^{5,9}$ were abundant in boreal species (*G. barretti*: 4 ± 3%; *G. atlantica*: 9%). Also 25-Me-$C_{26:2}\Delta^{5,9}$ ($i$-$C_{27:2}\Delta^{5,9}$), and 25-Me-$C_{26:2}\Delta^{9,19}$ ($i$-$C_{27:2}\Delta^{9,19}$) were produced by boreal species, representing together 3 ± 2% of total FAs in *G. barretti* and 5% in *G. atlantica*. Because these peaks co-eluted, the individual concentrations might represent isomeric mixtures. *G. hentscheli* possessed low amounts of (mid-)Me-$C_{26:2}\Delta^{5,9}$ and 25-Me-$C_{26:2}\Delta^{9,19}$ ($i$-$C_{27:2}\Delta^{9,19}$) (< 2%). Similarly, *S. rhaphidiophora* had low amounts of (mid-)Me-$C_{26:2}\Delta^{5,9}$ and ($a$)$i$-$C_{27:2}\Delta^{5,9}$ (together 2%, Table 1).

- **C$_{28}$**: *G. atlantica*, *G. barretti* and *G. parva* contained C$_{28:2}$ with $\Delta^{11,21}$ configuration, comprising 1.8 ± 1% of total FAs in *G. barretti*, 1.9% in *G. atlantica* and 3.6 ± 1.7% in *G. parva*. *S. rhaphidiophora* contained a low amount of C$_{28:2}\Delta^{5,9}$ (1.5 ± 0.4%) (Table 1).

- **C$_{29}$**: The dominant LCFA in *S. rhaphidiophora* was (mid)-Me-C$_{28:2}\Delta^{5,9}$ with a contribution of 5.5 ± 0.6% to total FAs (Table 1).

## Stable C isotope values (δ$^{13}$C)

Stable C isotope values (δ$^{13}$C) of dominant FAs ranged between -18 ‰ (95 percentile) and -26 ‰ (5 percentile) and showed similar patterns across all demosponges (Fig 4, Table 2). The δ$^{13}$C values of the dominant MBFAs, Me-C$_{16:0}$, Me-C$_{18:0}$, and also Me-C$_{18:1}\omega$12 (and $\omega$14) were enriched in $^{13}$C compared to other bacterial fatty acids, (*a(i)*-C$_{15:0}$, C$_{16:1}\omega$7, C$_{18:1}\omega$7) (Fig 4, Table 2). The most depleted FA was *i*-C$_{17:1}\omega$7 (-25.7 ± 1.3 ‰ δ$^{13}$C). The different LCFA isomers were analyzed as one, because isomers co-eluted or were at least not well separated on GC (Fig 2). However, we could assign separate isotope values for (*a*)*i*-C$_{27:2}$ and Me-C$_{26:2}$ (Fig 4). The LCFAs showed less isotopic variation compared to bacterial FAs, but still ranged between -25 and -19 ‰ (5–95 percentile) (Fig 4, Table 2). Me-C$_{26:2}$ and (*a*)*i*-C$_{27:2}$ had relatively similar δ$^{13}$C values, -20 and -21 ‰, in *G. barretti* and *G. atlantica*, but a more prominent difference of -21 and -24 ‰ was observed in the hydrogenated samples (*n* = 1 per species), indicating that peak overlap blurred the isotopic values.

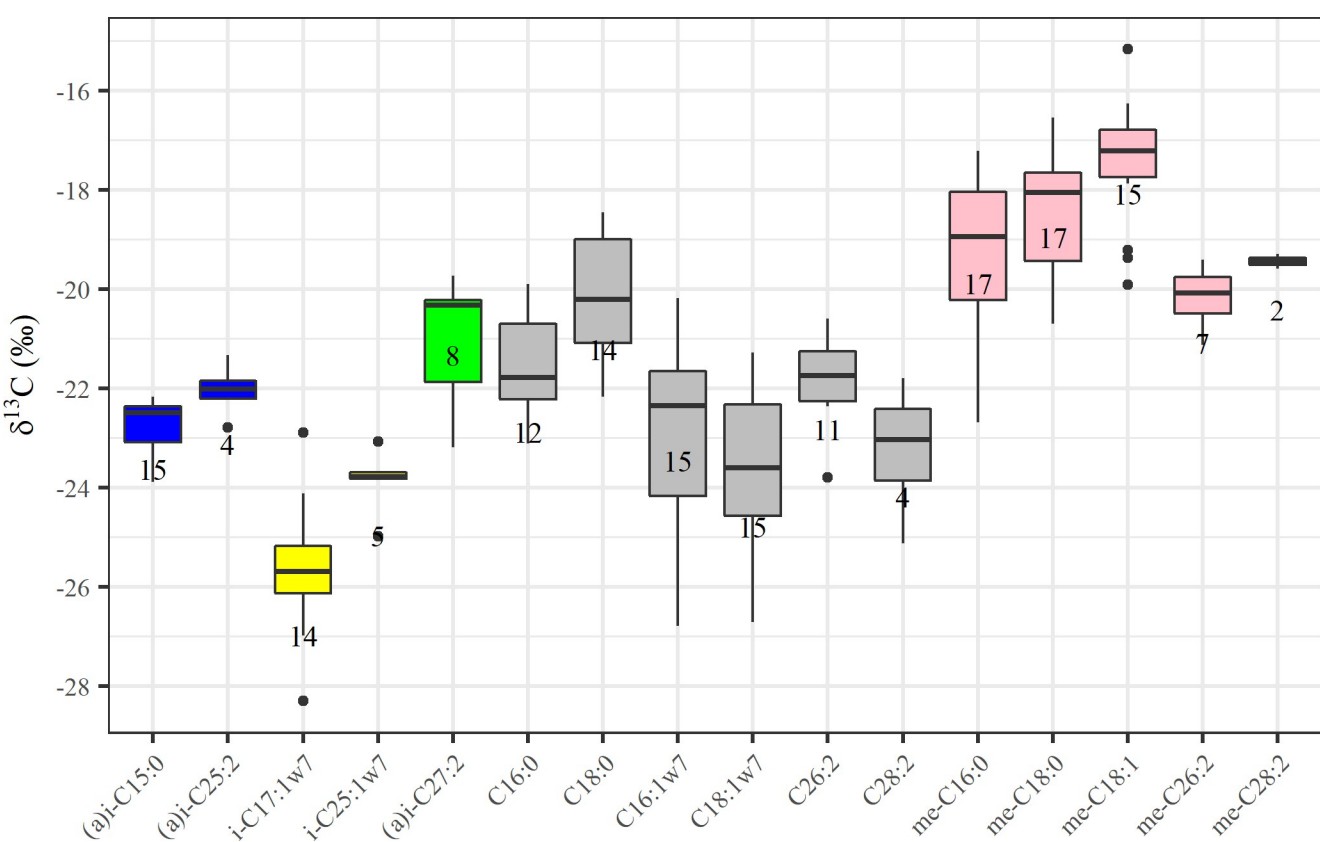

**Fig 4. δ$^{13}$C composition of precursors and dominant LCFAs in analyzed demosponges.** Sponge species were pooled together, and the median is indicated in the box plot as black line. The numbers depict the sample size (individual FAME samples). The colors are used to match bacterial precursor FAs with sponge-produced LCFAs. Pink is used for (mid-)Me-branched FAs, grey is used for linear FAs, blue and yellow indicate *(a)i*-branched FAs with distinct δ$^{13}$C that may end up in an isomeric mixture, indicated by green (see Fig 5 for biosynthetic pathways).

**Table 2. δ¹³C values (mean ± SD) of (bacterial) FA precursors and dominant LCFAs of all species combined.**

| Category | Fatty acid biomarker | Average $\delta^{13}C$ (‰) ± SD |
|---|---|---|
| $a(i)$-branched FA | $(a)i$-$C_{15:0}$ | -22.8 ± 0.6 |
| | $(a)i$-$C_{25:2}$ | -22.0 ± 0.6 |
| | $i$-$C_{17:1}\omega7$ | -25.7 ± 1.3 |
| | $i$- $C_{25:1}\omega7$ | -23.9 ± 0.7 |
| | $(a)i$-$C_{27:2}$ | -21.0 ± 1.2 |
| Linear FA | $C_{16:0}$ | -21.6 ± 1.1 |
| | $C_{18:0}$ | -20.2 ± 1.2 |
| | $C_{16:1}\omega7$ | -23.0 ± 2.0 |
| | $C_{18:1}\omega7$ | -23.6 ± 1.5 |
| | $C_{26:2}$ | -21.8 ± 0.9 |
| | $C_{28:2}$ | -23.2 ± 1.4 |
| Mid-branched FA | Me-$C_{16:0}$ | -19.3 ± 1.6 |
| | Me-$C_{18:0}$ | -18.4 ± 1.1 |
| | Me-$C_{18:1}$ | -17.4 ± 1.3 |
| | Me-$C_{26:2}$ | -20.2 ± 0.6 |
| | Me-$C_{28:2}$ | -19.4 ± 0.2 |

## Discussion

### Bacterial FAs

High concentrations of isomeric mixtures of MBFAs were found in all five sponge species analyzed, independent of species and location (Table 1). A predominance of MBFAs is considered to be a typical feature of Demospongiae, because it is not observed in any other organism, sediment or water [12,17,42]. Typical position of branching is between ω5 and ω9 [12], resulting in predominance of 8- and 9-Me-$C_{16:0}$ and 10- and 11-Me-$C_{18:0}$ in this study, in agreement with previously reported MBFAs [43,44]. MBFAs are typically produced by bacteria, so they are presumably made by distinctive and sponge-specific eubacterial symbionts. It has been hypothesized that these bacteria were widespread in the geological past and were inherited in the protective environment of distinctive sponge hosts in modern marine environments [8,12]. This hypothesis has been further supported by genomic analysis on *Geodia* sp. revealing similar microbial communities between species with little geographical variation [45].

A proposed candidate phylum for MBFAs is Poribacteria, a unique and abundant phylum in HMA sponges [46], since a positive relation between MBFA concentration and Poribacteria abundance was found across several sponge species [44]. Metagenome analyses showed that Poribacteria are a prominent phylum in *G. barretti* [47,48], but are rare or even absent in *G. hentscheli* [49], which shows a dominance of Acidobacteria, Chloroflexi, and Proteobacteria, phyla that are also abundant in *G. barretti* [47,48]. This suggests that either the MBFAs belong to one of the above-mentioned phyla, or that the MBFAs are shared among microbial phyla, as their chemotaxonomic resolution is lower compared to genomic analysis. In the environment, MBFAs are primarily found in nitrogen and sulfur reducers (chemoheterotrophs) and oxidizers (chemoautotrophs) that are mostly members of the (large) proteobacteria family [50–53]. Nitrogen and sulfur reduction and oxidation processes are conducted in deep-sea sponges such as *G. barretti* [26,54,55], and oxidation processes are coupled to $CO_2$ fixation, although associated $CO_2$ fixation is likely to contribute < 10% of the carbon demand of deep-sea sponges [56]. The poribacteria in sponges were also characterized as mixotrophic bacteria, able to fix $CO_2$ using the ancient Wood–Ljungdahl (reversed acetyl-CoA) pathway [57]. The

isotopic enrichment in MBFAs (Fig 4, Table 2), agrees with earlier observations for *G. barretti* [58], and might thus be linked to nitrogen and sulfur transforming processes and potentially $CO_2$ fixation. It will be interesting to perform an isotope-tracer study [56] with $^{13}C$-$CO_2$ to assess $CO_2$ incorporation in the abundant MBFAs, perhaps combined with nitrification (or sulfur oxidation) inhibitors, similar to Veuger et al. [59].

The most depleted FA ($i$-$C_{17:1}\omega7$, Fig 4, Table 2) is considered a chemotaxonomic marker for the sulfur reducing bacteria *Desulfovibrio* sp. [60]. The isotopic difference between $i$-$C_{17:1}\omega7$ and MBFAs suggest that these markers are not from the same microbial consortium. The more general bacterial markers (e.g. $(a)i$-$C_{15:0}$, typical of gram-positive bacteria and $C_{16:1}\omega7$ and $C_{18:1}\omega7$, typical of general gram-negative bacteria [15] had intermediate $\delta^{13}C$ values (Fig 4, Table 2). Such values can be the result of isotopic averages from different pathways, since they are more general bacterial markers, or they might represent general heterotrophy on organic matter with $\delta^{13}C$ value from -24 to -22 ‰ in the western Arctic [61].

The low contribution of FAs with a chain length of $C_{20}$ to $C_{24}$ typical of phytoplankton and zooplankton (e.g. $C_{20:5}\omega3$ and $C_{22:6}\omega3$) indicates that sinking zoo- and phytoplankton are not contributing much to sponge diet, at least not directly. These findings support increasing evidence that *G. barretti* (and other North-Atlantic deep-sea sponges) primarily feed on dissolved organic matter and pelagic and associated bacteria [62,63]. Part of the bacterial FAs might thus be originated from pelagic bacteria, rather than bacterial endosymbionts, although this contribution is expected to be low compared to the high number of bacterial symbionts ($10^{11}$ cells $mL_{sponge}^{-1}$, [62]). A higher contribution of phytoplankton markers in boreal *Geodia* spp. (*G. atlantica* and *G. barretti*) compared to Arctic species (Table 1) might be linked to water depth, as boreal species were sampled from ~300 m and Arctic species from ~600 m, while also environmental factors, such as permanent ice coverage (Langseth Ridge) and a generally lower primary production in the Arctic compared to the boreal North-Atlantic ocean [64] might play a major role.

The overall high abundance of bacterial FAs (56–79% of total FAs across all five analyzed deep-sea demosponge species, Fig 3) fits with their classification as HMA sponges and supports the idea that microbial endosymbionts play a pivotal role in sponge metabolism [2,3]. It is important to notice that the contribution of endosymbionts is likely even higher, since archaea are not detected with (PL)FA analysis [65], while they were also found to be abundant in *G. barretti* [47,48].

## Sponge LCFAs

Although bacterial FA profiles were very similar among the studied Tetractinellid species, the sponge-specific LCFA composition was more species-specific. The dominant unsaturation in LCFAs, was double unsaturation at $\Delta^{5,9}$ position in all species analyzed, which is typical feature of demosponges [5,10]. Similarly, the linear $C_{26:2}\Delta^{5,9}$, $(a)i$-$C_{25:2}\Delta^{5,9}$ and/or $i$-$C_{27:2}\Delta^{5,9}$, present in all species analyzed (Fig 5, Table 1), are common LCFAs of demosponges, (e.g. [17,39,66], for an overview see [8]). We found (mid-)Me-branched $\Delta^{5,9}$ LCFAs in all species, except *G. parva* (Fig 5, Table 1). Also, Thiel et al. [17] found them in *G. barretti* and some other Demospongiae (*Haliclona* sp., *Petrosia* sp.) but not in all analyzed Demospongiae. We also identified novel LCFAs: $i$-$C_{27:2}\Delta^{9,19}$ (in *G. atlantica*, *G. barretti*, and *G. hentscheli*) and $\Delta^{11,21}$ in $C_{26:2}$ and $C_{28:2}$ (*G. barretti* and *G. parva*). The presence of $\Delta^{11}$ unsaturation ($C_{26:2}\Delta^{11,21}$ and $C_{28:2}\Delta^{11,21}$), identified via DMDS derivatization, is uncommon in sponge LCFAs. Barnathan et al. [67] found $\Delta^{11}$ unsaturation in a series of monoenic FAs, including $C_{28:1}$, in a tropical demosponge species (order Axinellida), but no dienic LCFAs with $\Delta^{11}$ unsaturation have been described so far. The configuration indicates that $\Delta^{11}$ desaturase might be active in these species; however, the activity of this enzyme in sponges has not been reported.

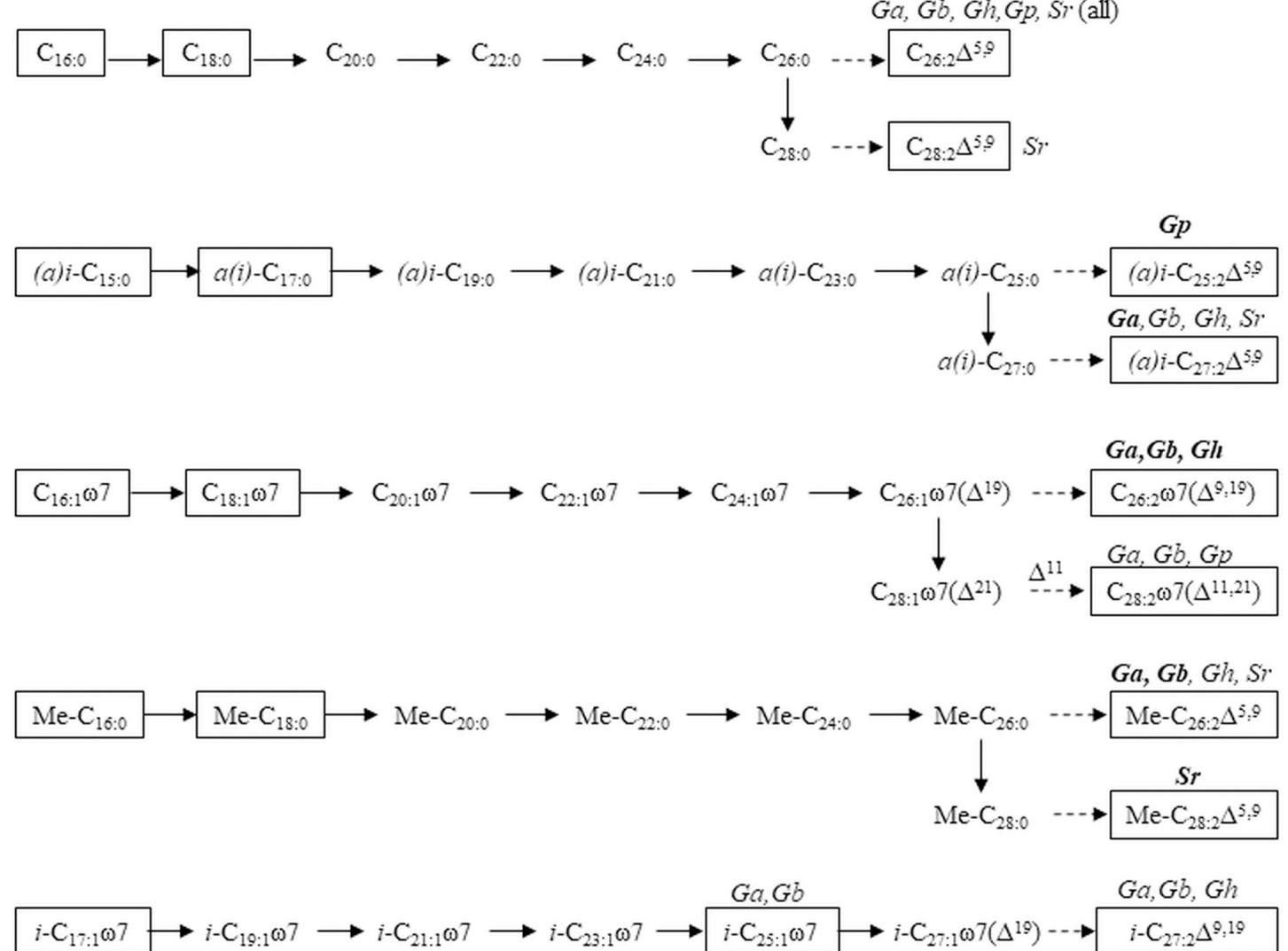

**Fig 5. Proposed biosynthetic pathways of (microbial) precursors to LCFA in examined Tetractinellid species.** FAs detected in the studied sponges are shown in a black rectangle, solid arrows indicate elongation, dashed arrows indicate $\Delta^{5,9}$ desaturation. The species encompassing each LCFA are indicated with abbreviated names as in Table 1 and names in bold means that the LCFA is dominant in that specific species.

*G. atlantica* and *G. barretti* had almost identical LCFA profiles (Table 1), suggesting that these species might be closely related, as was earlier suggested based on their sterol and amino acid composition [68], but deviates from molecular phylogeny that places them further apart [27]. The FA profile of *G. hentscheli* resembled those of *G. barretti* and *G. atlantica* and based on molecular phenology, *G. hentscheli* is a sister species of *G. barretti*. However, *G. parva* produced distinct LCFAs compared to the other *Geodia* spp., the *i*- and *a*-$C_{25:2}\Delta^{5,9}$ and this species is also phylogenetically apart from the other *Geodia* spp. [27]. A dominance of *(a)i*-branched $C_{25:2}$ has been found in another Geodiidae family (*Geodinella*) [69]. Finally, also *S. rhaphidiophora* produced a distinct LCFA, Me-$C_{28:2}\Delta^{5,9}$, a LCFA that has been described for demosponges of the family *Aplysinidae* [13,43].

Each of the three dominant Tetractinellids of Artic sponge grounds (*G. hentscheli, G. parva* and *S. rhaphidiophora)* produced distinct LCFAs (Table 1) that can serve as chemotaxonomic

markers. The morphology of these sponges is very similar, so LCFA analysis provides an additional method to identify each species. Furthermore, the distinct LCFAs could be useful as trophic markers to study the ecological role of deep-sea sponges in the environment. No geographical differences in LCFA composition of Arctic Tetractinellids were found (S1 Table) suggesting that the environment has a limited influence on the LCFA composition, which is a prerequisite for using LCFA as chemotaxonomic markers.

**Biosynthetic pathways of prominent sponge fatty acids.** The identification of branching in LCFAs allows identification of its short chain precursors and biosynthetic pathways. As demonstrated by various *in vivo* incorporation studies with radioactive substrates [16,39,66,70], sponges elongate FA precursors by adding 2 C atoms at the carboxylic acid end and desaturate at $\Delta^5$ and $\Delta^9$ (visualized in [10]), revealing $C_{16:0}$ as precursor for the common $C_{26:2}\Delta^{5,9}$, while $C_{16:1}\omega7$ was identified as precursors for $C_{26:2}\Delta^{9,19}$ (Fig 5). There is no evidence for branching to be introduced by sponges, so *i*- and *a*-$C_{15:0}$ were identified as precursors of *i*- and *a*-$C_{27:2}\Delta^{5,9}$ (Fig 5) [39], while Me-$C_{16:0}$ has been identified as precursor for Me-$C_{26:2}\Delta^{5,9}$ and Me-$C_{28:2}\Delta^{5,9}$ (Fig 5) [13,16,71]. Finally, we hypothesize that *i*-$C_{17:1}\omega7$ is the precursor for *i*-$C_{25:1}\omega7$ and *i*-$C_{27:2}\Delta^{9,19}$ found in *G. atlantica*, *G. barretti*, and *G. hentscheli* (Fig 5).

Application to the present study showed that most LCFAs could be linked to precursors via established pathways, with hypothetical intermediates since hardly any were found in detectable abundance (Fig 5). The C isotopic differences in bacterial precursors were (partially) reflected in C isotopic composition of LCFAs (Fig 4, Table 2), although the differences were not as prominent in LCFAs compared to their precursors. One explanation is that a mixture of C sources is used by the host to elongate precursors to LCFAs, while also methodological aspects might contribute. A (much) longer analytical column might help improving separation of LCFAs.

The schematization of Fig 5 shows the benefit of using both ω and Δ (and mixed) nomenclatures in sponge lipid research. Annotations from the terminal end (ω and *(a)i* notation) (Fig 1) are convenient to show biosynthetic pathways as these positions do not change with elongation (Fig 5). However, the typical $\Delta^{5,9}$ unsaturation is more convenient to show with Δ annotation, as an ω notation would alter with varying C chain length (Fig 5). Ambiguity arises in ω notation of methyl-branching, because *a(i)* notation is used for terminally branched FAs and describes total C atoms (including the methyl group(s)), while Me notation is used for MBFAs and describes the C number of the backbone (excluding methyl group (s)) and counts the position of the branching from the carboxylic acid end (and not the terminal (ω) end, Fig 1). This might lead to confusion about the total C number, which is needed to correct measured isotope values for the extra methyl group, and about the ω position of unsaturation (start counting from the end of the backbone, excluding the methyl-group) and the conversion from ω to Δ notation (Fig 1). We added this discussion to create awareness and would like to recommend including a description of the notation in the methods and presenting both nomenclature when a mixture of notation styles is used.

## Conclusions

In this study we identified FAs of prominent habitat-building demosponges (order Tetractinellida) from the boreal-Arctic deep Atlantic Ocean. All five species investigated contained predominantly bacterial FAs, in particular isomeric mixtures of MBFAs (Me-$C_{16:0}$ and Me-$C_{18:0}$) (together >20% of total FAs). The MBFAs were isotopically enriched compared to linear and *(ante)iso*-branched FAs. The sponge-produced LCFAs with chain lengths of $C_{24}$-$C_{28}$ were linear, mid- and *a(i)*-branched and had predominantly the typical $\Delta^{5,9}$ saturation. They also produced (yet undescribed) branched and linear LCFAs with $\Delta^{9,19}$ and $\Delta^{11,21}$ unsaturation,

namely $i$-$C_{27:2}\Delta^{9,19}$, $C_{26:2}\Delta^{11,21}$, and $C_{28:2}\Delta^{11,21}$. *G. parva* and *S. rhaphidiophora* each produced distinct LCFAs, while *G. atlantica*, *G. barretti*, and *G. hentscheli* had a similar LCFA profile, although each species had different predominant ones. The typical FA profiles of North-Atlantic deep-sea demosponges can be used as chemotaxonomic and trophic markers. We proposed biosynthetic pathways for dominant LCFAs from their bacterial precursors, which were supported by small isotopic differences in LCFAs that support the idea that sponges acquire building blocks from their endosymbiotic bacteria.

## Supporting information

**S1 Fig.** Mass spectra of DMDS conducts of $C_{26}$ (a,b) and $C_{28}$ (c,d) LCFA with $\Delta^{9,19}$ (a,c) and $\Delta^{11,21}$ (b,d) unsaturation.
(PDF)

**S1 Table. All fatty acid compositional data.** This excel file contains fatty acid data (µg g DW$^{-1}$ and relative abundance (%), in PL and TL) of individual specimens. The excel file also contains the fragments of Me-branched $C_{16}$ and $C_{18}$, the relative positions of saturated (branched and linear) FAMEs in hydrogenated samples and the isotope data.
(XLSX)

## Acknowledgments

We thank Antje Boetius for supporting and promoting this study and organizing the PS101. We thank the captain and crew of PS101 for excellent support at sea. We thank late Hans Tore Rapp (UiB) for organizing the G.O. Sars expeditions and excellent project coordination. We thank Desmond Eefting for analytical assistance. We thank master students Gydo Geijer, Sean Hoetjes, David Lankes, Floor Wille, and Femke van Dam for their help in the lab and with analyses. We thank Eva de Rijke, Samira Absalah and Stefan Schouten for their help in protocol development. Irene Rijpstra and Volker Thiel are acknowledged for their help with FA identification. We thank Pieter van Rijswijk and Marco Houtekamer for sharing their analytical knowledge and identification libraries. Paco Cardenas is acknowledged for sharing his taxonomic knowledge. We thank Gilles Barnathan and an anonymous reviewer for their constructive review.

## Author Contributions

**Conceptualization:** Anna de Kluijver.

**Data curation:** Anna de Kluijver.

**Formal analysis:** Anna de Kluijver, Klaas G. J. Nierop, Teresa M. Morganti.

**Funding acquisition:** Jasper M. de Goeij, Furu Mienis, Jack J. Middelburg.

**Investigation:** Anna de Kluijver, Klaas G. J. Nierop, Martijn C. Bart.

**Methodology:** Anna de Kluijver, Klaas G. J. Nierop, Teresa M. Morganti, Jasper M. de Goeij.

**Resources:** Martijn C. Bart, Beate M. Slaby, Ulrike Hanz, Jasper M. de Goeij, Furu Mienis, Jack J. Middelburg.

**Validation:** Anna de Kluijver, Klaas G. J. Nierop.

**Visualization:** Anna de Kluijver.

**Writing – original draft:** Anna de Kluijver.

**Writing – review & editing:** Anna de Kluijver, Klaas G. J. Nierop, Teresa M. Morganti, Martijn C. Bart, Beate M. Slaby, Ulrike Hanz, Jasper M. de Goeij, Furu Mienis, Jack J. Middelburg.

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
