## [Decision Letter · Decision Letter 0]

9 Nov 2020

PONE-D-20-31393

Bacterial precursors and unsaturated long-chain fatty acids are biomarkers of North-Atlantic demosponges

PLOS ONE

Dear Dr. de Kluijver,

Thank you for submitting your manuscript to PLOS ONE. After careful consideration, we feel that it has merit but does not fully meet PLOS ONE’s publication criteria as it currently stands. Therefore, we invite you to submit a revised version of the manuscript that addresses the points raised during the review process.

We look forward to receiving your revised manuscript.

Kind regards,

Clara F. Rodrigues

Academic Editor

PLOS ONE

Journal Requirements:

3. We noted in your submission details that a portion of your manuscript may have been presented or published elsewhere:

'The fatty acid profile of one of the five sponge species that we analyzed has been used in an experiment with isotopically labelled food sources. The incorporation of label (13C) into fatty acids was used to quantify uptake by bacterial symbionts and sponge host. The manuscript is attached.'

Please clarify whether this publication was peer-reviewed and formally published.

If this work was previously peer-reviewed and published, in the cover letter please provide the reason that this work does not constitute dual publication and should be included in the current manuscript.

Additional Editor Comments:

Dear Dr de Kluijver

Thank you for submitting your manuscript entitled "Bacterial precursors and unsaturated long-chain fatty acids are biomarkers of North-Atlantic demosponges" to PLOS ONE. We have received two reviews for this manuscript. Both reviewers considered this a very well written interesting manuscript. I recommend the acceptation of this manuscript after minor revisions

Reviewers' comments:

Reviewer's Responses to Questions

**Comments to the Author**

1. Is the manuscript technically sound, and do the data support the conclusions?

Reviewer #1: Yes

Reviewer #2: Yes

2. Has the statistical analysis been performed appropriately and rigorously? 

Reviewer #1: Yes

Reviewer #2: N/A

3. Have the authors made all data underlying the findings in their manuscript fully available?

Reviewer #1: Yes

Reviewer #2: Yes

4. Is the manuscript presented in an intelligible fashion and written in standard English?

Reviewer #1: Yes

Reviewer #2: Yes

5. Review Comments to the Author

Reviewer #1: Reviewer’s comments

The manuscript of Dr. de Kluijver and co-authors contains results of a solid piece of work and it is clearly written. It is of great interest according to several points of view.

Firstly, the Authors investigated deep-sea sponges, collected in the North-Atlantic Ocean during different scientific expeditions, for phospholipid fatty acid (FA) compositions. Four sponges belonging to the genus Geodia, another one to the genus Stelletta, were studied. It should be noted that fatty acids of Geodia sponges are lowly documented to date.

Secondly, these sponge species being classified as high microbial abundance species, the Authors determined phospholipid FA composition for typically bacterial FAs and typically sponge FAs. Bacterial FAs (branched and monoenic short-chain FAs) constituted the majority of total FAs in all five deep-sea demosponge species. Interestingly, high concentrations of mid-chain branched FAs (MBFAs) were found in all five sponge species analyzed. A predominance of MBFAs is considered to be a typical feature of Demospongiae and it is also known that MBFAs are typically produced by bacteria.

Thirdly, biosynthetic pathways for some long chain FAs from their possible bacterial precursors are proposed. These pathways are supported by the small isotopic differences found in LCFAs using the δ13C values obtained by GC-C-IRMS.

Fourthly, the Authors conducted an interesting discussion according to a chemotaxonomic point of view about the possible origins of bacterial symbionts used as metabolic sources by sponges.

The use of DMDS adducts is very appropriate and clearly presented in the text and in figures.

In conclusion, the scientific investigation sounds fine and the strategic approach and fatty acid analyses are appropriated conducting to great results.

Thus, this paper should be recommended to be published with minor revision.

Minor corrections and improvements

Title

I would suggest to write “North-Atlantic deep-sea sponges“, although this is noted in the Short Title and Key words.

Methods

Sponge collection

This part is well detailed and interesting but a precious indication is missing. Indeed, it is important to give details about the voucher specimens and in which scientific institutions  they were deposited. This is usually required and it seems particularly important since the sponges studied are deep-sea ones. Furthermore, they mainly belong to the genus Geodia, which is relatively less documented.

FAME analysis

GC analysis. Split or splitless ?

Results

It seems surprising that FA compositions of total lipids and phospholipids were found similar. This point should be discussed. Thus, an interesting information is missing. The percentages of total lipids relative to dry matter are given but not the ones of phospholipids relative to total lipids. It seems necessary in order to a better comprehension of data showed in Table S1

Reviewer #2: This manuscript provides a detailed characterisation of fatty acid composition in a number of sponges. The manuscript is very well written, experimental methods are clear and so are the results. I find that the manuscript can be published almost as it is. I have just a few question, which may be considered.

p. 3 and elsewhere. The authors introduce and discuss FA from bacterial endosymbionts in relation to the extracted fatty acids and as precursors for long chain fatty acids. Sponges are known to retain particles less than 1 µm in size and may therefore feed on suspended bacterial cells, so what about a potential contribution of fatty acids obtained by the sponges from digestion of bacterial prey (maybe the discussion, Line 387-389 could be expanded)?

Line 420. I suggest it is specified that it is the LSFAs that may be used as phylogenetic markers, if this is the case? I was a bit confused first time I read this part.

6. PLOS authors have the option to publish the peer review history of their article (what does this mean?). If published, this will include your full peer review and any attached files.

Reviewer #1: **Yes: **prof. Barnathan gilles

Reviewer #2: No

---

## [Author Response · Author response to Decision Letter 0]

15 Dec 2020

Editorial comments

Permits: No permits were required for field site access in international waters. No special permits were required for Norwegian research expeditions in the Barents Sea region. Both statements have been added to the methodology section.

Protocol: The protocol is available online at protocols.io: (dx.doi.org/10.17504/protocols.io.bhnpj5dn)

Dual publication. This submission presents new data that are not part of the publication by Bart et al. (2020). This submission presents the identification, and natural abundance (isotopic) composition of fatty acid profiles of five different demosponge species. For one single species, namely G. barretti, we have used the fatty acid profile presented in this submission to study ecological performance (presented in Bart et al. (2020)). As Bart et al. (2020) is a follow-up experimental study, the natural background data for G. barretti are presented in this submission. Bart et al. (2020) presents incorporation of deliberately added 13C in specific fatty acids to assess the role of symbiotic bacteria in food uptake with no mention of natural abundance composition. Because of the different scope and research question, there is no overlap in tables and figures between our submission and Bart et al., (2020). The profiles of the other 4 species have not been reported before nor are part of Bart et al. (2020). 

Reviewer 1

Title: this is a good suggestion, we added “ deep-sea” to the title

Sponge collection: Indeed, the precious information on voucher specimens deposition has been added.

Fame analysis: We used cold-on-column injection, this has been added.

Results: It may not be surprising that the fatty acid (FA) compositions of the phospholipids (PL) fraction and of total lipids are fairly similar, as the majority of fatty acids is present as phospholipids (our study; e.g. Parzanini et al., 2019), although we are aware that glycolipids are also found in sponges (e.g. Bergé and Barnathan, 2005). We cannot exclude that our PL fraction also contained other lipids, such as glycolipids, as was shown by (Heinzelmann et al., 2014). We agree that comparison of fatty acid composition between different lipid classes is interesting, but this was beyond the scope of our study, so we did not investigate this sufficient to justify any conclusions and a comprehensive discussion. However, we added that most fatty acids are present as phospholipids with a reference (Parzanini et al., 2019) and added the quantified PLFA concentrations.

Reviewer 2

Discussion: Indeed, part of the bacterial fatty acids might be derived from bacterial prey, although their contribution is probably low (Leys et al., 2018). We added this to the discussion.

We also changed FAs to LCFAs in L420 (now L427), as we indeed discuss LFCAs as taxonomic markers.

---

## [Editor Report · Decision Letter 1]

18 Dec 2020

Bacterial precursors and unsaturated long-chain fatty acids are biomarkers of North-Atlantic deep-sea demosponges

PONE-D-20-31393R1

Dear Dr. de Kluijver,

Thank you for addressing all the issues/suggestions raised by the reviewers.

We’re pleased to inform you that your manuscript has been judged scientifically suitable for publication and will be formally accepted for publication once it meets all outstanding technical requirements.

Kind regards and Merry Christmas

Clara F. Rodrigues

Academic Editor

PLOS ONE
---

## [Editor Report · Acceptance letter]

6 Jan 2021

PONE-D-20-31393R1 

Bacterial precursors and unsaturated long-chain fatty acids are
biomarkers of North-Atlantic deep-sea demosponges 

Dear Dr. de Kluijver:

I'm pleased to inform you that your manuscript has been deemed suitable for publication in PLOS ONE. Congratulations! Your manuscript is now with our production department. 

Kind regards, 

on behalf of

Dr. Clara F. Rodrigues 

Academic Editor

PLOS ONE